# Statistical Analysis of Plasma Dynamics in Gyrokinetic Simulations of Stellarator Turbulence

**DOI:** 10.3390/e25060942

**Published:** 2023-06-15

**Authors:** Aristeides D. Papadopoulos, Johan Anderson, Eun-jin Kim, Michail Mavridis, Heinz Isliker

**Affiliations:** 1School of Electrical and Computer Engineering, National Technical University of Athens, 157 80 Athens, Greece; 2Department of Space, Earth and Environment, Chalmers University of Technology, SE-412 96 Göteborg, Sweden; 3Centre for Fluid & Complex Systems, Coventry University, Coventry CV1 5FB, UK; ejk92122@gmail.com; 4Department of Physics, Aristotle University of Thessaloniki, 541 24 Thessaloniki, Greece

**Keywords:** gyrokinetic simulations, drift waves, stochastic theory, information geometry, time series analysis, 52.35.Ra, 52.25.Fi, 52.35.Mw, 52.25.Xz

## Abstract

A geometrical method for assessing stochastic processes in plasma turbulence is investigated in this study. The thermodynamic length methodology allows using a Riemannian metric on the phase space; thus, distances between thermodynamic states can be computed. It constitutes a geometric methodology to understand stochastic processes involved in, e.g., order–disorder transitions, where a sudden increase in distance is expected. We consider gyrokinetic simulations of ion-temperature-gradient (ITG)-mode-driven turbulence in the core region of the stellarator W7-X with realistic quasi-isodynamic topologies. In gyrokinetic plasma turbulence simulations, avalanches, e.g., of heat and particles, are often found, and in this work, a novel method for detection is investigated. This new method combines the singular spectrum analysis algorithm with a hierarchical clustering method such that the time series is decomposed into two parts: useful physical information and noise. The informative component of the time series is used for the calculation of the Hurst exponent, the information length, and the dynamic time. Based on these measures, the physical properties of the time series are revealed.

## 1. Introduction

Experimental investigations of heat and particle transport in magnetically confined (MC) plasmas, such as those found in tokamaks and stellarators, are often found to be elevated compared to what is predicted by collisional predictions [1]. The optimization of the magnetic field in stellarators was shown to significantly reduce their neo-classical losses. This makes ion-temperature-gradient (ITG)- and trapped-electron-mode (TEM)-driven turbulence [1] among the main physical mechanisms responsible for the increased transport levels in the core stellarator plasma. Furthermore, outside of the bulk plasma, large-scale events termed blobs and avaloids may be generated that have a significant impact on the overall transport [2,3,4,5,6,7,8,9,10,11]. Moreover, there exist modes that connect the bulk and edge plasma directly, namely avalanches and streamers. Other types of filaments, such as edge-localized modes (ELM), may also have a harmful impact on the first wall [9]. The understanding and characterization of these structures has been elusive due to their intermittent nature and complicated generation process. Hence, it is of great interest to improve the understanding and predictive capability of large-scale structures in MC plasmas.

Many different transport models and codes have been developed to study these effects in different detail and aspects. In this work, we utilize the gyrokinetic model, which is among the most promising tools for the study of micro-scale turbulence and its interaction with large-scale modes, such as the mitigating effects of zonal flows and the detrimental effects of streamers and avalanches. Here, we numerically evaluate the generated quasi-stationary time series of the surface-averaged heat flux from the gyrokinetic simulations of ITG-mode-dominated plasma with adiabatic electrons [1] performed by the GENE code [12]. These simulations are carried out in order to model turbulence in the core of the stellarator W7-X at different locations and with varying temperature and density gradients using a realistic and optimized quasi-isodynamic topology [13].

The analysis of the results is performed by means of singular spectrum analysis (SSA) and information length estimation. The SSA methodology [14] is a well-known mathematical method for analyzing the different components of time traces, where deterministic components are filtered out. In weakly stochastic systems, oscillatory components, such as normal modes, that are typically superposed on the turbulent signal may conceal interesting turbulent dynamics and have to be removed before analysis. The SSA method has rendered acclaim in widely different areas, such as geology [15], economics [16], and medicine [17]; however, a couple of interesting instances in plasma physics are to be found [18,19]. In Ref. [19], an analysis using SSA that proved valuable for the detection of intermittent events, such as blobs, in experimental signals or code simulations was presented. Understanding and characterizing the effects of large-scale structures is, e.g., possible by employing a probabilistic approach where the probability density function (PDF) is computed. Although there is a relatively small likelihood of these intermittent events, they may be spatially extended and have a large amplitude; thus, the total mediated transport may be significant. In terms of the PDFs, these events dominate and elevate the tails of the distribution function. Assessing for temporal changes in the system-state PDF is one option for characterizing intermittent events. Imposing a metric for the thermodynamic length allows one to measure the distance that a system travels between thermodynamic equilibrium states, as described by a PDF [20,21,22,23,24,25,26,27,28]. This is a novel methodology to measure distance in statistical space. When a PDF continuously changes with time, the information length measures the total number of different statistical states that a system passes through in time [26,27,28].

The work performed in this paper extends the previously described situation by combining an analysis using the SSA methodology and the possible identification of events in the time traces, the information length *L*, with a discussion on the statistical properties of the time traces, such as variability and persistence.

This paper is organized as follows. The gyrokinetic model and setup of the stellarator simulations are discussed in Section 2, and the statistical analysis and its interpretation are explained in Section 3 and Section 4, respectively. The paper is concluded with a discussion in Section 5.

## 2. GK Model and Simulation Set-Up

In this section, the GENE gyrokinetic code used to obtain the time series data is briefly described, and the physical parameter setup in the simulations are presented. Turbulence produced by plasma micro-instabilities is considered the main reason for the anomalous transport observed in fusion devices. Small-scale plasma turbulence, meaning fluctuations of physical observables that present spatial scales in the order of the Larmor radius and frequencies significantly smaller compared to the cyclotron frequency (gyro-frequency), is studied by gyrokinetic theory [29]. Many numerical codes have been created to simulate gyrokinetic turbulence in fusion machines. In this work, we used the GENE (gyrokinetic electromagnetic numerical experiment), which is a Eulerian δf code developed by the Gene Development Team and is publicly available [30]. A detailed description of the equations solved by the code, as well as their numerical implementation and the produced output of the code, can be found in Refs. [12,31,32,33,34,35].

We performed gyrokinetic, nonlinear, and flux tube (local) simulations of ITG-mode-driven turbulence using the realistic magnetic geometry of the W7-X stellarator. Our specific aim was the modelling and statistical analysis of turbulent heat fluxes in a realistic configuration of quasi-isodynamicity. The stellarator’s magnetic geometry that was used in these simulations was created with the use of the VMEC code, which calculated the configuration of the MHD equilibrium. The latter served as input for the GIST code, which calculated the various geometric and physical quantities needed for the gyrokinetic equations while transforming them in the coordinate system that GENE uses; see, e.g., [36]. We performed non-linear flux tube simulations at two different radial positions. The GIST code used the toroidal flux as a flux surface label, which was normalized to be in the range between zero and unity. Thus, *s* is defined as
(1)$$s=\frac{\Phi }{\Phi_{edge}}\;\mathrm{with}\;s\in \left[0,\;1\right]$$
with Φ being the toroidal flux and $\Phi_{edge}$ being the toroidal flux of the last closed flux surface. Simulations were performed with s=0.5,0.81. GENE uses a field-aligned coordinate system that takes advantage of the differences in the characteristics of the turbulence perpendicular to and along the magnetic field lines. Assuming that x,y, and *z* express these field-aligned coordinates, *x* is considered the radial normalized coordinate, which is defined as $x=\rho_{N}=\sqrt{s}$. Applying the latter in the studied test cases, the values of the radial positions of the flux tubes simulated were $x_{0}=0.7,\;0.9$, respectively, to the mentioned *s* values. The *z* coordinate was along the magnetic field line and parameterized it. In this case, z=θ, where θ is the poloidal Boozer angle ranging in both simulated flux tubes between z=θ∈[−π,π]. The variation of the value of the normalized magnetic field along the *z* coordinate for both tubes is presented as calculated by the GIST code in Figure 1.

Finally, *y* represents the binormal coordinate, which is defined in GENE as
(2)$$y=\frac{\sqrt{s_{0}}}{q_{0}}\left(q\theta -\zeta \right)=\frac{\sqrt{s_{0}}}{q_{0}}\alpha ,$$
where the index 0 denotes flux functions calculated on the selected surface, which are treated as constants. The quantities θ and ζ are the poloidal and toroidal Boozer angles, respectively, $q_{0}$ represents the safety factor value at the radial position $x_{0}=\sqrt{s_{0}}$ of the flux tube, *q* represents the safety factor function, and α=qθ−ζ represents a magnetic line label. In these simulations, both tubes started from the outboard plane with α=0.

In the flux tube approximation, a small tube was created along the magnetic field line. The equilibrium temperature, density, and pressure were kept constant for the tube; their parallel dependence was, therefore, neglected, and they were completely determined by their value on the field line and their radial gradient. In the direction perpendicular to the magnetic field line, for the *x* and *y* coordinate directions, periodic boundary conditions were assumed, where $L_{x}$ and $L_{y}$ were the lengths of the simulation domain in the corresponding directions creating the tube along the line in the *z* direction. For both simulated radii, two different temperature gradients were examined: $a/L_{T}=1.0,\;4.0$, with a=0.512 m being the minor radius of the W7-X stellarator. The minor radius was also used as a reference length for normalization purposes. All simulations were considered electrostatic, meaning that fluctuations of the magnetic field are not considered. Simulations that were extended in time were needed to reach a stationary phase in the evolution of the physical observables in the time series. For the numerical grid resolution in the *x*, *y*, and *z*
GENE magnetic coordinates, we used typical values for the number of grid points for such (local) simulations, with $n_{x}=60$, $n_{y}=60$, and $n_{z}=160$. The flux tube of these local simulations had a radial length $L_{x}=137.129$ in units of the ions’ Larmor radius $\rho_{i}$, and the minimum wave number used in the simulations in the binormal direction *y* was $k_{y\min}=0.05$, which was used by the code to determine the binormal $L_{y}$ length of the tube.

In gyrokinetic theory, the fast gyromotion is averaged out through transformations that remove the dependence of the equations from the gyro-angle used for describing the cyclotron motion of particles. The latter results in the so-called gyrokinetic Vlasov equation, which describes the evolution of the perturbating part of each particle species distribution function in a five-dimensional space with three spatial and two velocity coordinates instead of the usual six dimensions. The remaining two coordinates of the velocity space are the velocity component parallel to the magnetic field, $u_{\|}$, and the magnetic moment μ, which is related to the component of the velocity perpendicular to the magnetic field. The number of grid points used in discretizing the velocity space for the parallel velocity $u_{\|}$ and the magnetic moment μ are, respectively, $n_{v}=40$ and $n_{w}=20$, where the symbols $u_{\|},\mu ,\to v,$ and *w* are used. In the velocity space, the extent of the direction of the parallel velocity is $l_{v}=3.0$ in the units of the thermal velocity, while the length of the simulation grid in the direction of the magnetic moment μ is $l_{w}=9.0$ in units of $T_{0\mathrm{ref}}/B_{\mathrm{ref}}$, with $T_{0\mathrm{ref}}$ being the value of the ions’ temperature at equilibrium and $B_{\mathrm{ref}}$ being the reference value of the magnetic field strength provided by the GIST magnetic geometry file. The value of the plasma beta is assumed to be close to zero for both tubes. Finally, for the tube at the radial position $x_{0}=0.7$, the safety factor is equal to $q_{0}=1.114$, and the magnetic shear is ${\hat{s}}_{0}=-0.145$, while for the tube at the radial position $x_{0}=0.9$, the safety factor is equal to $q_{0}=1.066$, and the magnetic shear is ${\hat{s}}_{0}=-0.23$. Moreover, the density gradient is set to $a/L_{n}=0,\;1$, and the electrons are considered adiabatic.

We examined the time series of the ions’ radial heat flux calculated by volume averaging the normalized velocity space moment of the fluctuating part of their distribution function $F_{1i}\left(x,u\right)$, which can be defined as
(3)$$Q_{i}\left(x\right)=\;\int \int \int \;F_{1i}\left(x,u\right)v_{\xi }^{x}\frac{m_{i}u^{2}}{2}d^{3}u$$
where $\frac{m_{i}u^{2}}{2}$ is the ion kinetic energy and $v_{\xi }^{x}$ is the contravariant component of the generalized E×B velocity used for taking the radial projection of the heat flux. Concerning the units of the various physical observables, one must take into account that GENE uses the gyrokinetic equations in their dimensionless form; thus, its calculated quantities are also dimensionless, meaning that normalization is used throughout the code implementation. We selected as reference length $L_{ref}$ the minor radius of the stellarator *a*, as already mentioned. Due to the latter, time was measured in units of $a/c_{i}$, with $c_{i}$ the ions’ sound speed under radial heat flux was measured in gyro-Bohm units; see Figure 2. GENE produces time series with a varying time step selected during the simulation for optimized accuracy. The statistical tools used demanded data equally spaced in time; thus, the time series was interpolated with the use of cubic splines and re-evaluated at equidistant data points in time without changing the simulated time interval or the number of data points evaluated.

## 3. Statistical Methods

The output generated by the GENE gyrokinetic code forms a weakly steady-state time series with complex features due to the non-linearly interacting plasma modes. The structural behavior of the stochastic time series can be analyzed by the singular spectrum analysis (SSA) method [37]. Inspired by Ref. [19], where the SSA was used for the statistical analysis of intermittent transport events in the tokamak scrape-off layer, the SSA method is used here for the analysis of the heat flux (Q) time series provided by the nonlinear gyrokinetic code GENE. Compared to the SSA method of Ref. [19], the SSA is implemented here in conjunction with a clustering method. In this way, the time series is decomposed to its physical components, as will be described later on. In general, the SSA method accepts a time series in the form $X=\left(x_{1},x_{2},\ldots ,x_{N}\right)$ as input and yields a decomposition of X into identifiable subseries components, $X=X_{1}+X_{2}+\ldots +X_{m}$. SSA consists of four steps: embedding, decomposition, grouping, and reconstruction. The embedding step involves the linear map $T_{SSA}:$$T_{SSA}\left(X\right)=\tilde{X}$, where $\tilde{X}$ is defined as the trajectory matrix (a Hankel matrix where the entries are equal on the anti-diagonals). $\tilde{X}$ is of size L×K, where K=N−L+1 and *L* is a window length (lag) specified based on the application at hand [37]. The *i*-th column of the trajectory matrix is of the form $X_{i}={\left(x_{i},x_{i+1},\ldots ,x_{i+L-1}\right)}^{T}$. Next, the trajectory matrix $\tilde{X}$ is decomposed into a sum of rank-1 components as $\tilde{X}={\tilde{X}}_{1}+\ldots +{\tilde{X}}_{d}$, ${\tilde{X}}_{i}=\sigma_{i}U_{i}V_{i}^{T}$, where $U_{i}$ and $V_{i}$ are the *L* and *K* dimensional vectors of the unit norm, respectively, and $\sigma_{i}$ are non-negative numbers (singular values). The above decomposition is the singular value decomposition (SVD) of ${\tilde{X}}_{i}$, and $d=\max\left\{j:\sigma_{j}>0\right\}=rank\left({\tilde{X}}_{i}\right)$. A grouping strategy is applied to the rank-1 components of the trajectory matrix $\tilde{X}$. If it is assumed that the index set I={1,…,d} is partitioned into *m* disjoint sets $I_{1},\ldots I_{m}$, the grouped matrix decomposition of the trajectory matrix is $\tilde{X}={\tilde{X}}_{I_{1}}+\ldots +{\tilde{X}}_{I_{m}}$, where $I_{k}=\left\{i_{1},\ldots ,i_{p\left(k\right)}\right\}\subset I$ and ${\tilde{X}}_{I_{k}}={\tilde{X}}_{i_{1,k}}+\ldots +{\tilde{X}}_{i_{p,k}}.,k=1\ldots m$. The partition of *I* into *m* disjoint sets is achieved using hierarchical clustering based on the weighted series distance matrix $d_{w}=1-\rho_{w}$, where $\rho_{w}\left(i,j\right)\equiv {\left(X_{i},X_{j}\right)}_{w}/\left({∥X_{i}∥}_{w}{∥X_{j}∥}_{w}\right),{\left(X,Y\right)}_{w}\equiv \sum_{l=1}^{N}w_{l}X_{l}Y_{l}$, $w_{l}=card\left(\{x\left(l\right)\in \tilde{X}\}\right)$, ${∥X∥}_{w}=\sqrt{{\left(X,X\right)}_{w}}$, and $X_{i}$ and $X_{j}$ are reconstructed series obtained from the corresponding trajectory matrices ${\tilde{X}}_{i}$ and ${\tilde{X}}_{j}$.

Hierarchical clustering (HC) is a clustering method that produces a hierarchy of clusters. In this work, agglomerative HC is considered, which is a bottom-up approach in the sense that each observation forms its own cluster and observations are then merged in an additive manner as we move up the hierarchy. Each observation is one of the reconstructed series $X_{i}$, i=1,…,d. The distance between observations *i*, *j* is given by ${\left(d_{w}\right)}_{i,j}$. In the first HC step, pairs of observations *i*, *j* form a cluster if $i=\mathrm{agr}\min_{i^{*}}{\left(d_{w}\right)}_{i^{*},j}$. In the second HC step, since clusters have been formed, there must be a definition of cluster distance. This is provided by the linkage clustering method, for which there are many choices, including single-linkage, complete-linkage, average-linkage, and Ward-linkage clustering. The single-linkage clustering method is used here, where the clusters *A* and *B* are at a distance $\min\left({\left(d_{w}\right)}_{i,j},i\in A,j\in B\right)$. The HC steps continue with upwards clustering until the desirable number of clusters, nc, are formed (the HC stopping criterion in this work) or until the clusters are too far apart to merge. In Figure 3, the SSA method with the HC automatic grouping of the eigenvalues is tested on a signal $y=x+\sin\left(2\pi x\right)+n$ containing white Gaussian noise *n* (with a signal-to-noise ratio of SNR = 10 dB), a trend, and oscillation. Running the SSA with HC where three clusters of eigenvalues are created results in the correct identification of the three subseries corresponding to noise, trend, and oscillation.

Finally, in the reconstruction step on each of the trajectory matrices ${\tilde{X}}_{I_{k}}$ of the grouped matrix decomposition $\tilde{X}={\tilde{X}}_{I_{1}}+\ldots +{\tilde{X}}_{I_{m}}$, a projection operator $P_{H}$ is applied, which is just antidiagonal averaging followed by the inverse embedding transform $T^{-1}$. It is proven [37] that the reconstructed series $X_{I_{k}}$ is optimal in the sense that it minimizes ${∥{\tilde{X}}_{I_{k}}-T\left(X_{I_{k}}\right)∥}_{F}$, where ${∥M∥}_{F}$ is the Frobenius norm of the matrix M. From SSA, we obtain a time subseries of $X_{i}$, where i=1,…,d are the SSA components, and it is possible to calculate their dynamic time and information length. The dynamic time $\tau \left(t\right)$ is a time scale over which the probability of $X_{i}$ changes, on average, at time *t* (denoted as $p\left(X_{i},t\right)$). The probability $p\left(X_{i},t\right)$ is estimated from a subset (window) of $X_{i}$ samples of length WL produced around the time index *t*. In particular, first, the samples of $X_{i}$ are interpolated to equally spaced time instances (separated by dt); then, from the samples of each window *w* of size WL, the $p\left(X_{i},t\right)$ is calculated at the time instant *t* at the middle of the window’s time interval. Then, the window *w* is moved (running window) by one time sample, and $p\left(X_{i},t+dt\right)$ is calculated at the time instant t+dt, and so on. The calculation of $p\left(X_{i},t\right)$ is carried out based on the histogram of *w* samples. Usually the histogram-produced $p\left(X_{i},t\right)$ is non-smooth and a smoothing Gaussian kernel is applied. In particular, the probability *p* is approximated by
(4)$$\hat{p}\left(y\right)=\frac{1}{WL\cdot h_{K}}\sum_{m=1,i_{m}\in w}^{WL}K\left(\frac{y-X_{i_{m}}}{h_{K}}\right),$$
where $K\left(x\right)$ is a positive function (the kernel) and $h_{K}$ is the bandwidth parameter. A Gaussian kernel (standard normal pdf) and an optimum bandwidth $h_{K}^{*}$ (for the Gaussian kernel) are chosen as [38]
(5)$$h_{K}^{*}={\left(\frac{2}{3WL}\right)}^{1/5}\min\left({\hat{\sigma }}_{R},{i\hat{q}r}_{R}\right),$$
where ${\hat{\sigma }}_{R}$ and ${i\hat{q}r}_{R}$ are the empirical standard deviation and interquartile deviation, respectively. ${\hat{\sigma }}_{R}$ is optimal in the sense that it minimizes the $\ell^{2}$ approximation error ${∥p-\hat{p}∥}_{2}$.

Then, $\tau_{i}\left(t\right)$ of the $X_{i}$ subsequence is given by [28]
(6)$$\tau_{i}{\left(t\right)}^{2}=1/\int dX_{i}\frac{1}{p\left(X_{i},t\right)}{\left(\frac{\partial p\left(X_{i},t\right)}{\partial t}\right)}^{2}.$$

From the dynamic time, the information length, $L_{i}\left(t\right)$, can be directly calculated [28]:(7)$$L_{i}\left(t\right)=\int_{0}^{t}ds\frac{1}{\tau_{i}\left(s\right)}.$$

In the implementation of Equations (Equation 6) and (Equation 7), since $X_{i}$ and *t* are discrete, the integrals and differentiations become summations and differences, respectively.

Another quantity of interest is the Hurst exponent [39] of X. In general, 0<H<1 but if H=0.5, the series is considered random (uncorrelated); if H>0.5, the series has a long-term positive autocorrelation, meaning that high (low) values in the series X will have a higher probability of being followed by another high (low) value. Conversely, if H<0.5, in the long run, with high probability, high (low) values in X will have a higher probability of being followed by another low (high) value. The Hurst exponent is calculated by the rescaled range (RS) method as the exponent *H* such that $E\left[R\left(n\right)/S\left(n\right)\right]=Cn^{H}$ for n→∞, where *C* is a constant, $E\left[x\right]$ is the expected mean, $S\left(n\right)$ is the standard deviation of the series $X_{1},X_{2},\ldots ,X_{n}$, and $R\left(n\right)$ is the range of the *n* cumulative deviations from the mean; that is, $R\left(n\right)=\max\left(Z_{1},Z_{s},\ldots ,Z_{n}\right)-\min\left(Z_{1},Z_{s},\ldots ,Z_{n}\right)$, $Z_{j}=\sum_{i=1}^{j}\left(X_{i}-m\right)$, $m=\left(\sum_{i=0}^{n}X_{i}\right)/n$. Then, *H* is calculated as the slope of the line that fits the $\log\left(R(n)/S(n)\right)$ data as a function of $\log\left(n\right)$.

## 4. Results and Discussion

All data obtained from the GENE code are based on the variation of three parameters: the normalized radius of the stellarator s∈{0.5,0.8}, the normalized temperature gradient $R/L_{T}\in \left\{1,4\right\}$, and the normalized density gradient $R/L_{n}\in \left\{0,1\right\}$. Note that the threshold for instability is non-linearly dependent on both density and temperature gradients such that, with a linear increase in the density gradient, the corresponding growth rate changes in a non-linear way; see, e.g., Ref. [40]. Using the fluid model, the critical gradient is $R/L_{T}=2.7$ for $R/L_{n}=0$ and $R/L_{T}=3.0$ for $R/L_{n}=1.0$ in the local limit ($k_{\left|\right|}=0$). This indicates that the realizations with $R/L_{T}=1$ are just below the stability limit, whereas $R/L_{T}=4$ is in the unstable regime. In addition, the data have been normalized to zero mean and unit variance. In the following table, the Hurst exponent is shown for all 8 realizations of the parameters *s*, $R/L_{T}$, and $R/L_{n}$ for the complete or full-time series including oscillating and noise parts.

It is noted that cases $\left(0.8,1,1\right)$, $\left(0.5,1,1\right)$ are of less importance since the heat flux *Q* is close to zero and not in a quasi-stationary state and is thus difficult to compare with the other cases. Note that although the cases with higher density gradients yielded almost zero flux in comparison with the lower density gradient case, this may be indicated from the linear dynamics, where the driving is a competition between the density and temperature gradients. These almost-zero heat flux cases are neglected in the following analysis. Note that the information length in the zero heat flux cases are several orders of magnitude larger than the cases discussed here. Furthermore, all cases of interest in Table 1 exhibit Hurst exponents larger than 0.5, which is an indication of a positive autocorrelation in the time series, as discussed previously. In short, a Hurst exponent of 0.5 indicates that the signal is uncorrelated and thus mostly dependent on random events; however, if H>0.5, as is the case here, there is a persistence in the time series such that there is a higher probability of having repeated similar values. It is also of interest to discuss the effect of oscillatory components. A pure sinusoidal component is an example of deterministic dynamics that would have a Hurst exponent H=1.0 for time lags much smaller than the period. The values obtained here, [0.57,0.70], is in the same range as those found in Ref. [41] by the Langmuir probe measurement of turbulent fluctuations in edge plasmas, which were estimated to be in the range of 0.62 to 0.72 as measured in a collection of widely different devices such as tokamaks, stellarators, and one reverse field pinch (RFP). It is known that competing low-frequency, large-scale phenomena have significant impacts on the heat flux. These are primarily attributed to mitigating zonal flows and detrimental avalanches, which would both lead to increased Hurst exponents. A general observation regarding the time traces is that the comparable signal-to-noise ratio is vastly different for the different parameters, thus influencing the Hurst exponent. This will be discussed further below.

Applying the Wiener-–Khinchine theorem, the second-order structure function is the Fourier-transformed power spectrum. There is a relationship between the fractality of the process and the Hurst exponent, which yields a power spectrum of the form
(8)$$S\left(\omega \right)\propto \omega^{-\beta },$$
where β=2H−1 if the time series is indeed self-similar. This indicates a weak power-law scaling, which is in the same range as those indicated in gyrokinetic simulations of ITG turbulence in Ref. [42].

In Figure 4, the information length (*L*), computed using Equation (Equation 7), is shown for different values of the parameters $\left(s,R/L_{T},R/L_{n}\right)$. For the calculation of *L*, as described previously, a number of subseries samples, WL, of the running window were used. The subseries $X_{i}$ used corresponds to the non-noise part of the series *X*, as identified by the HC method. In particular, $X_{i}=\sum_{j=1,j\in nn}^{\left|nn\right|}X_{j}$, where the set nn contains the subseries indices belonging to the class of noise-free $X_{j}$ subseries. Initially, these samples were used for the calculation of the probability $p\left(X_{i},t\right)$ at sample time *t* in the middle of the running window. It is observed that the information length is monotonously increasing due to the positive definiteness inherent in its definition. It is thus pertinent to point out that the information length describes any change in the system (as defined by a change in the PDF of the system) with an increasing function; a system in a steady state would have a constant PDF and thus a constant information length (see Equation (Equation 7)). It is found, as expected, that the system is close to a quasi-steady state with an almost linearly or stepwise linearly increasing information length over time. However, in order to compute the PDFs, a finite number of sampling points must be used; thus, the accuracy of the PDF is dependent on the time and sampling points to estimate the PDF. Moreover, the sampling time cannot be too long since then rapid changes in the dynamics will not be captured. Thus, several tests with varying sampling points and window length were performed; see Figure 3 for a subset of these tests.

In Figure 5, the 1/τ parameter is presented as a function of the time samples for different values of the Hurst exponent (*H*), as given in Table 1 for the corresponding parameters $\left(s,R/L_{T},R/L_{n}\right)$. It is observed that peak values of the 1/τ parameters occur at the same time samples where the information length changes abruptly. This is to be expected since the information length is the integral of 1/τ, as seen in Equation (Equation 7). However, only consistent or persistent sampling over the time scale in the value of 1/τ in the integration will make significant changes in the information length. This persistence in the time series is a property that is related to the Hurst exponent.

When WL increases, a more accurate estimation of $p\left(X_{i},t\right)$ and consequently of *L* is obtained. However, as WL increases, the sampling length for the PDF results in a decreased number of time intervals or instances, *t* (which is N−WL), where the information length *L* is determined. It is observed in Figure 4 that as WL increases roughly by a factor of 5 from Figure 4a,b, the values of *L* decrease by roughly a factor of 20, while from Figure 4b,c, it can be seen that *L* decreases by a factor of 3 when WL increases by a factor of 2. Considering also that as WL increases, more time samples of the time trace are used, *L* is more accurately estimated for increasing values of WL, and convergent results are to be expected for WL=201. Fewer data points for the estimation of the PDF will result in situations where higher moments are not well-represented and the integration approximated by summation of the PDFs may become questionable. Since the dynamic time is related to the information length, it was also calculated for WL=201.

In analyzing the effect of increased density gradients ($R/L_{n}$), it is pertinent to remember that, as mentioned previously, there is a threshold value for turbulence. A change in linear stability is indicative of changes in the plasma state, whichis here approximated by the PDFs; e.g., in Figure 4c, the information length approaches an almost stepwise linear function in the cases of $R/L_{n}=1$. A linear increase in information is found in linear and non-linear Fokker–Planck models with linear time evolution. Note that several combinations of density and temperature gradients indicate conditions close to marginal stability where the dynamical system is expected to rapidly change between different states, including the effects of large-scale modes such as zonal flows. Due to this, the influence of the normalized radius and density and temperature gradients on information length will not be a linear relationship; however, the triplet (*s*, $R/L_{T}$, and $R/L_{n}$) will determine the path in the phase space and yield a Hurst exponent depending on the persistence or randomness of the resulting fluctuations. In particular, it is interesting to observe if there is a relationship between the final value of the information length according to Equation (Equation 7) and the triplet, as summarized in Table 2.

It is found that, for s=0.5, an increased density gradient $R/L_{n}$ in combination with a higher temperature gradient $R/L_{T}=4$ yield increased information length. However, keeping the density gradient constant ($R/L_{n}=0$) results in decreased information length. In the case with s=0.8, the increased density gradient results in decreased information length. Furthermore, it is observed that an increase in *s* gives an increased information length for both temperature gradients, except for the case with $R/L_{T}=4$ and $R/L_{n}=1$. One other interesting property of the time trace is the fractality, which seems to have an impact, as indicated by the varying Hurst exponent. Arranging the maximum information lengths with increasing Hurst exponents indicates that randomness and persistence in the time series give smaller values of information length. This could be explained by the closeness to marginal stability; the case with a Hurst exponent of 0.70 is close and may be impacted by non-linearly generated flows, yielding an increased Hurst exponent and a smaller information length. On the other hand, the smallest Hurst exponent is close to 0.5, indicating more randomness. This case is further from marginal stability, and the time trace is thus more prone to randomly generated transport events.

In Figure 6, Figure 7, Figure 8, Figure 9, Figure 10 and Figure 11a, it is observed that the case $R/L_{T}=4$ seems to mainly consist of the oscillatory component, whereas the random part is small compared to the case of $R/L_{T}=1$. This property seems to hold independently of the *s* and $R/L_{n}$ values. It has been suggested that the dynamic time can indicate large changes in the plasma state since it is the instantaneous distance between two PDFs describing the state. Although the information length is monotonously increasing, there exist large fluctuations in the dynamic time for all window lengths WL, indicating rapid changes in the plasma state. However, to have significant changes in the dynamics, a consistent trend is needed, as can also be seen in the information length. Thus, by analyzing the information and the dynamic time in tandem, a change in dynamics can be indicated. There are a few examples visible in the information length for WL=201 and (*s*, $R/L_{T}$, $R/L_{n}$) = (0.8, 1, 0) around a time step of 700–800. In this case, there is also a consistent minimum in the dynamic time, yielding, on average, a larger contribution (see Figure 10) to the information length since the dynamic time is inversely proportional to the information length. In Figure 7, there is a consistent minimum around time step 1500, which coincides with a rapid change in information. Quantification of the noise and oscillatory parts of the time trace is needed to delve deeper in the dependence of the Hurst exponent and the persistence of the properties of the oscillatory and noise parts since it was noted that it seems that the different cases have largely varying degrees of stochastic and oscillatory parts that may influence the information length and the Hurst exponent.

In Table 3, the signal-to-noise ratio (dB)is computed as SNR $=10\times \log_{10}\left(\frac{P_{\mathrm{signal}}}{P_{\mathrm{noise}}}\right)$ (where $P_{i}$ denotes the power), or variances of the oscillatory and noise parts of the time trace are presented. It is interesting to compare the time traces in Figure 7 and Figure 10 with the highest and lowest signal-to-noise ratios with the same gradients. It is observed that in Figure 7, the noise is only a very small part of the total time trace, whereas in Figure 10, the noise is a significant part of the signal. The information length is widely different (18.4 in Case 2 and 43.4 in Case 5), whereas the Hurst exponent is very similar (0.5687 in Case 2 and 0.5851 in Case 5). In comparing the cases with the largest Hurst exponents, such as Case 1 (0.7039) and Case 4 (0.6954), where these cases also have the same gradients, the signal-to-noise ratio is in the middle range. However, the information length increases from Case 1 (41.3) to Case 4 (55.8), which is due to the different position *s*. Note that in comparing Cases 3 and 6, it seems that these are quite similar in the sense that they are identified to have not too different Hurst exponents and information length in the mid range with similar signal-to-noise ratios. There are differences, in particular, in the information length, which is likely to be due to different conditions in the core and edge.

This indicates that it is possible to identify time instances where relatively sudden changes in the plasma state happen by investigations of the dynamic time and information length. However, unless the PDFs are investigated, there is no classification of what type of event it is. It is merely an indicator that the state is changing rapidly, which might be due to the generation of a high-transport event, such as an avalanche. If it is some other type of mode, it is not visible in the information length, and it is encoded in the PDF itself.

## 5. Discussion

In this work, time series generated by gyro-kinetic (GENE) simulations were investigated using the information geometric paradigm. The aim of this investigation was to quantify if changes in turbulent states could be identified by changes in information. As a testbed for the statistical analysis, we performed gyrokinetic simulations of ITG-mode-driven turbulence at two different magnetic flux radii, s=0.5 and s=0.8, with adiabatic electrons, which were relevant for the experimental conditions of the W7-X stellarator. The modeling and statistical analysis of turbulent heat fluxes in realistic configurations of different degrees of quasi-isodynamicity were the specific aims of the current paper. In order to create a reasonable test bed for the analysis, variations in density and temperature gradients were included in the test cases; however, only electrostatic perturbations were considered. The statistical analysis required long time series, where the statistics in the quasi-stationary state of the physical observables in the time series were of interest. These long time series (in terms of many turn-over times of the turbulence) were needed to capture the higher moments of the probability distributions (PDFs).

In this paper, we investigated the time series data from these simulations using the Hurst exponent, cluster analysis, and the information length approach. The Hurst exponent provided information on the long-term memory of the time series and was calculated by the RS method. The time series were further analyzed by the SSA method. In particular, the series were decomposed into subseries corresponding to the eigenvalues of the trajectory matrix, as described in the SSA. These eigenvalues were classified by the hierarchical clustering algorithm, which was built upon an appropriate weighted distance measure of the corresponding subseries. The eigenvalues were classified into three groups: noise, trends, and oscillatory parts. The part of the time series where noise was subtracted was then used to calculate the dynamic time and the information length.

A summary of the most important findings in this paper is presented in the following list.

The instantaneous change in information length is stored in the dynamic time, indicating that this could be a measure of instantaneous change in the plasma state. However, the dynamic time exhibits rapid fluctuations, and to have a consistent change in dynamics, a significant change in the information over a short interval is needed due to the effects of inertia on the dynamics.We find several distinct changes in state, from a quiescent state to a state with increased transport where large-scale transport events occur and are caused by the plasma turbulence in W7-X. In particular, the Hurst exponent (H) is consistently above H>0.5, suggesting that in this case, larger transport events can indeed occur. This methodology provides another tool in finding coherent transport events mediating large fluxes.By analyzing the information and the dynamic time in tandem, a change in dynamics can be indicated. Here, it is pertinent to remember that due to inertial effects, to have a significant change in the dynamics, a consistent change in the dynamic time is needed, as is then indicated in a rapid increase in information. It is only by the analysis of the original time trace that the properties of persistence can be elucidated, e.g., observing the fluctuations and oscillatory parts.

This study paves the way for further work on analyzing different types of plasma and investigating the possible usefulness of the information regarding the change of plasma state. However, it would be even more interesting to see if there are trends or possibilities in predicting changes in the dynamic time from a previous knowledge of the system, such as by statistical or specific types of machine learning techniques. Then, sudden changes in dynamics could be predicted by looking at series of PDFs describing the system. In particular, to validate the methodology, analyzing other time series data from simulation or experiments is necessary.

## Figures and Tables

**Figure 1 entropy-25-00942-f001:**
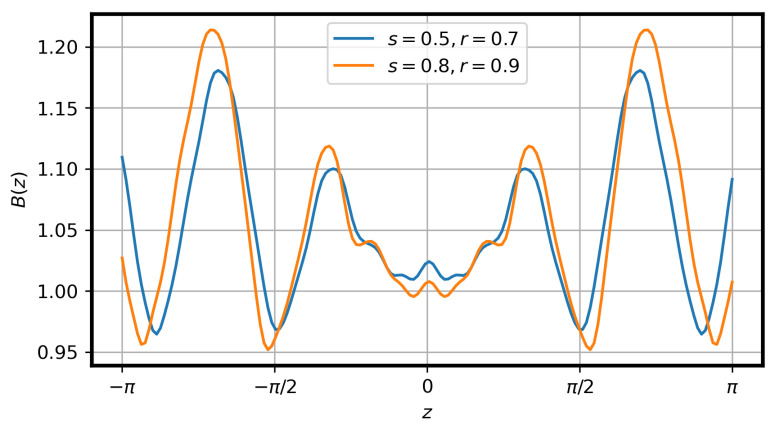
Variation of the value of the normalized magnetic field *B* with respect to *z* coordinate as defined for GENE for both flux tubes.

**Figure 2 entropy-25-00942-f002:**
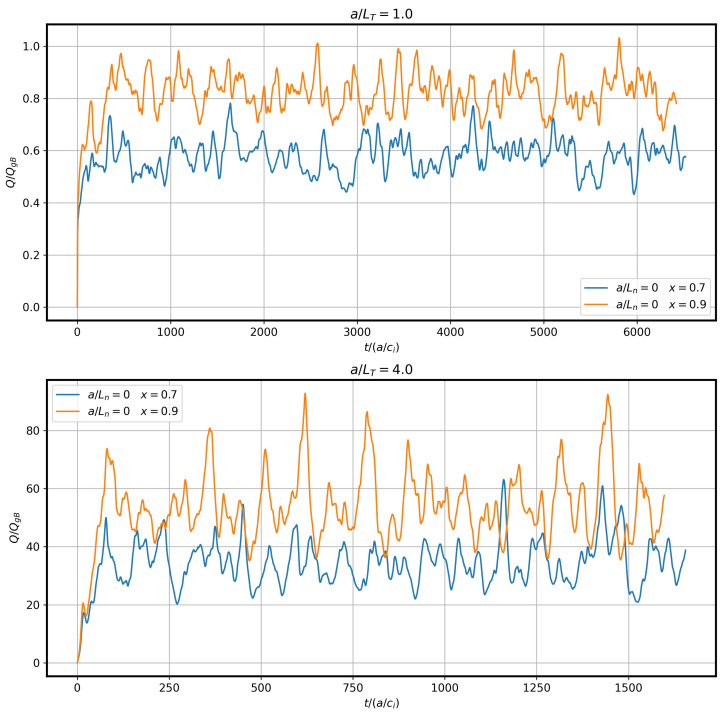
Ion heat flux $Q_{i}$ in gyro-Bohm units $Q_{gB}$ for the cases without density gradients.

**Figure 3 entropy-25-00942-f003:**
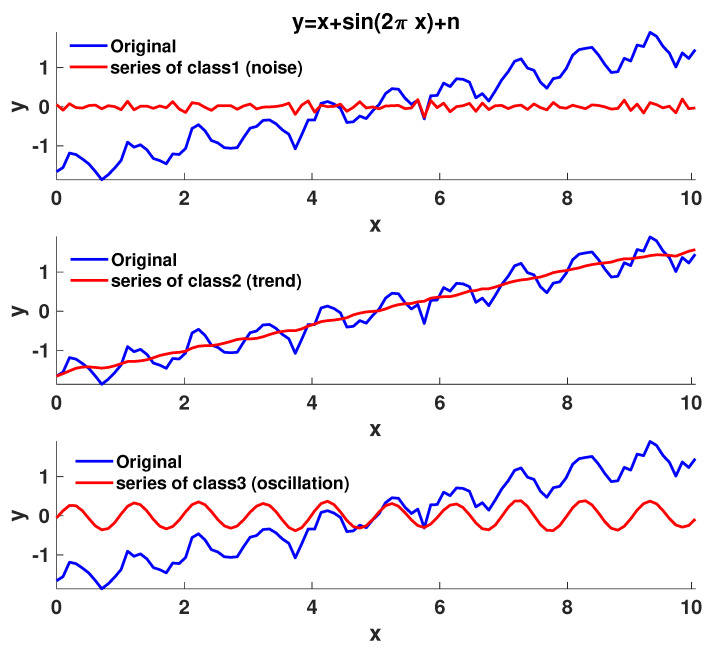
Validation of the SSA method by applying it on the time series *y*, where $y=x+\sin\left(2\pi x\right)+n$ and *n* is white Gaussian noise. Obviously, *y* contains a noise part, a trend part, and an oscillatory part. All three components are correctly identified by the SSA method.

**Figure 4 entropy-25-00942-f004:**
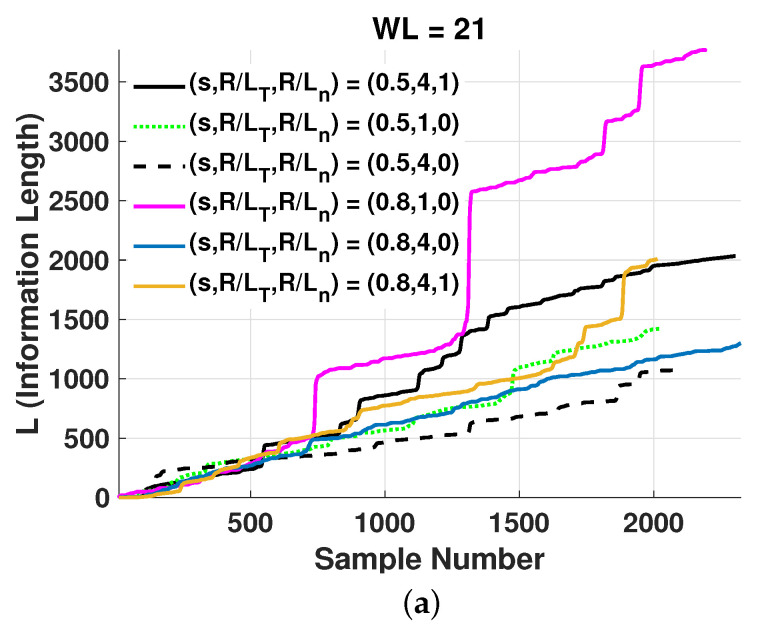
Information length (*L*) as a function of time samples, where running windows of samples have been used with different lengths WL (**a**) for WL=21, (**b**) for WL=101, and (**c**) for WL=201.

**Figure 5 entropy-25-00942-f005:**
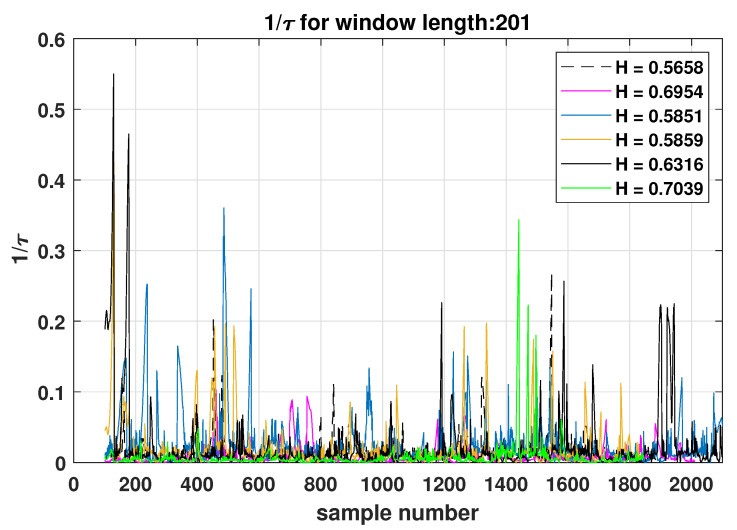
1/τ as a function of time samples for various values of the Hurst exponent *H* of Table 1.

**Figure 6 entropy-25-00942-f006:**
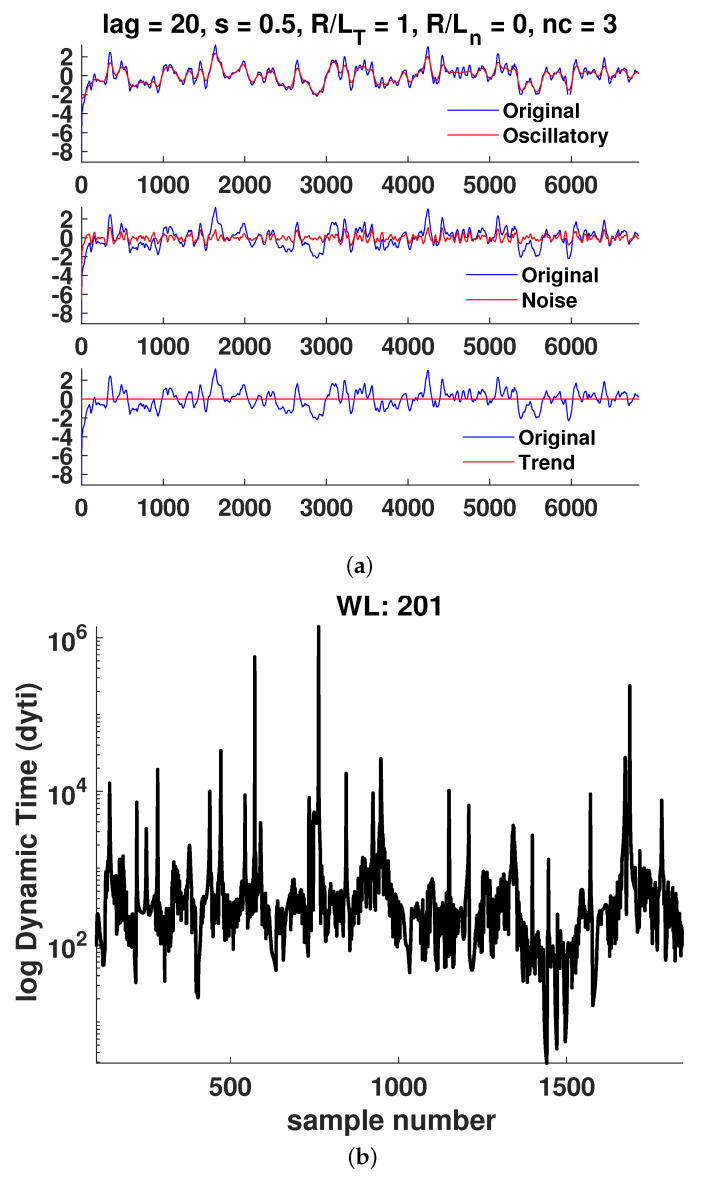
Case 1: (**a**) Hierarchical clustering results and (**b**) dynamic time calculations for WL=201. Here, nc is the maximal number of clusters, as defined in Section 3.

**Figure 7 entropy-25-00942-f007:**
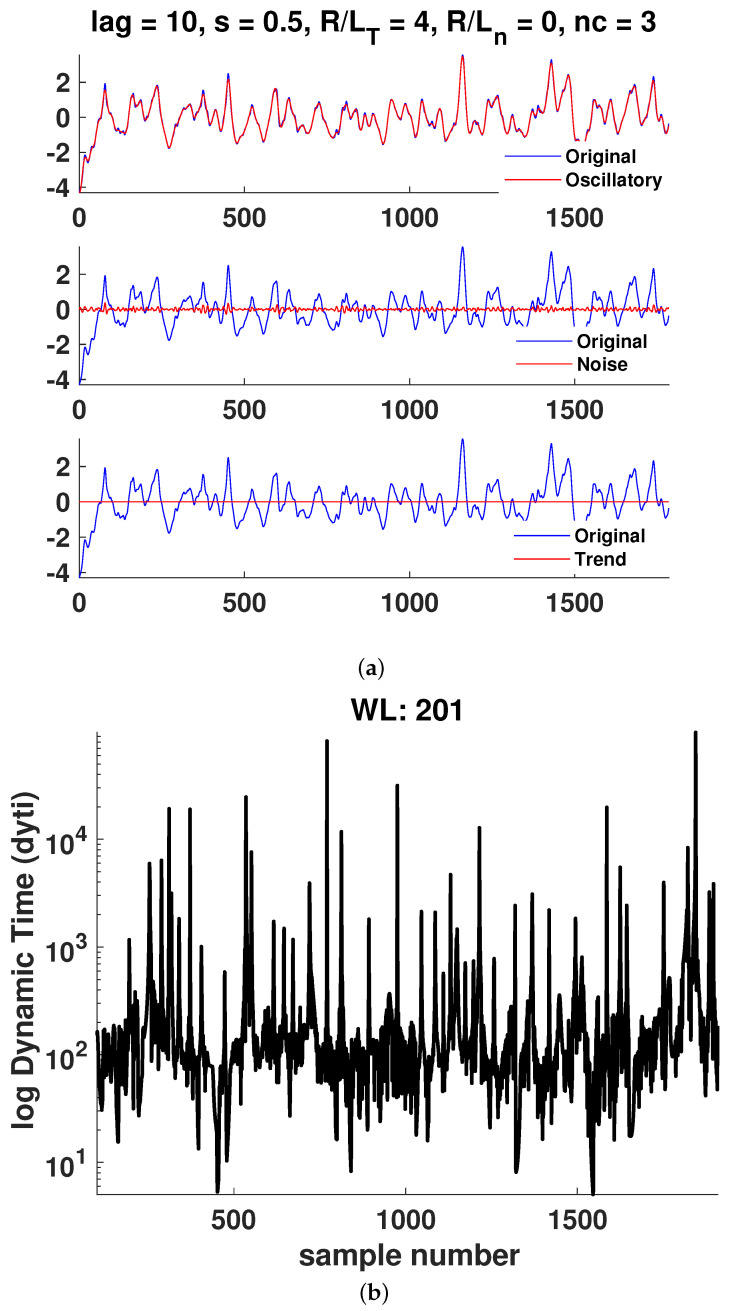
Case 2: (**a**) Hierarchical clustering results and (**b**) dynamic time calculations for WL=201. Here, nc is the maximal number of clusters, as defined in Section 3.

**Figure 8 entropy-25-00942-f008:**
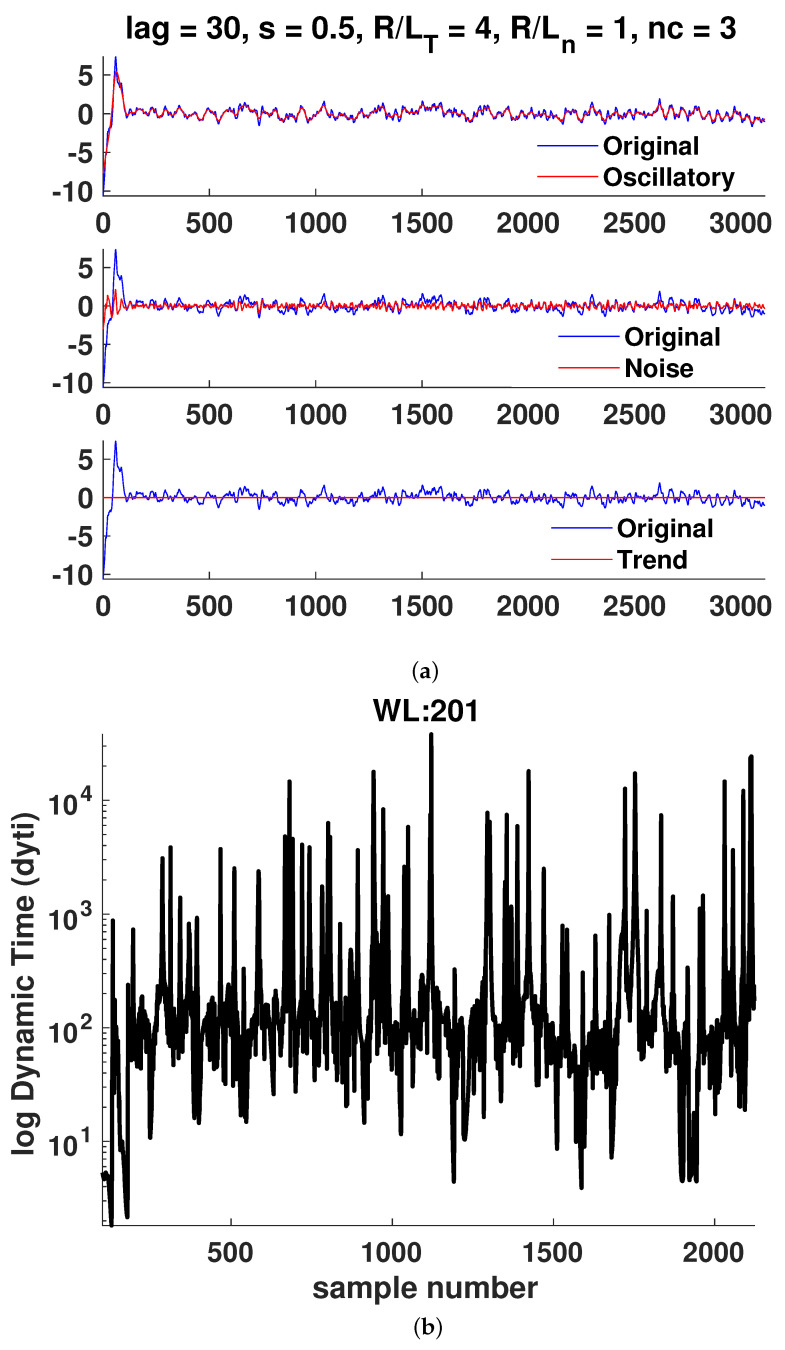
Case 3: (**a**) Hierarchical clustering results and (**b**) dynamic time calculation for WL=201. Here, nc is the maximal number of clusters, as defined in Section 3.

**Figure 9 entropy-25-00942-f009:**
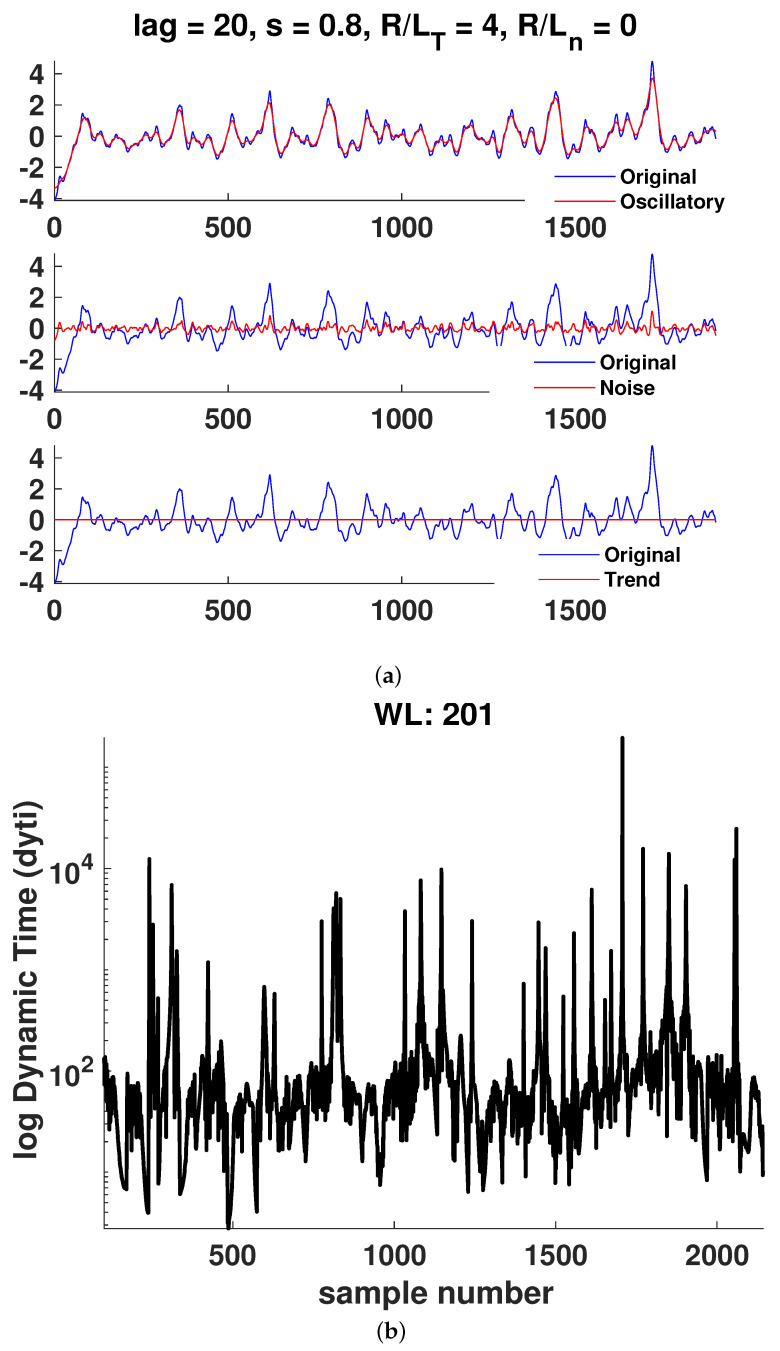
Case 4: (**a**) Hierarchical clustering results and (**b**) dynamic time calculations for WL=201. Here, nc is the maximal number of clusters, as defined in Section 3.

**Figure 10 entropy-25-00942-f010:**
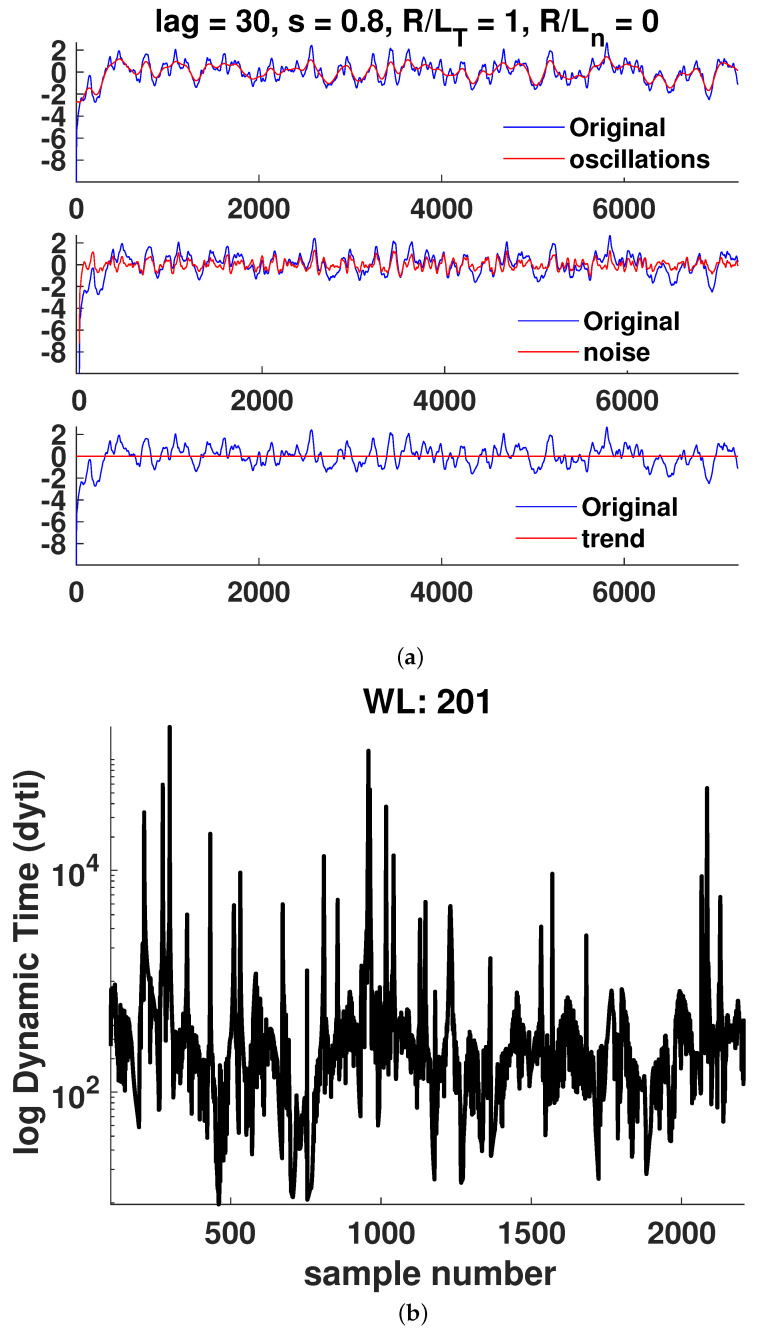
Case 5: (**a**) Hierarchical clustering results and (**b**) dynamic time calculations for WL=201. Here, nc is the maximal number of clusters, as defined in Section 3.

**Figure 11 entropy-25-00942-f011:**
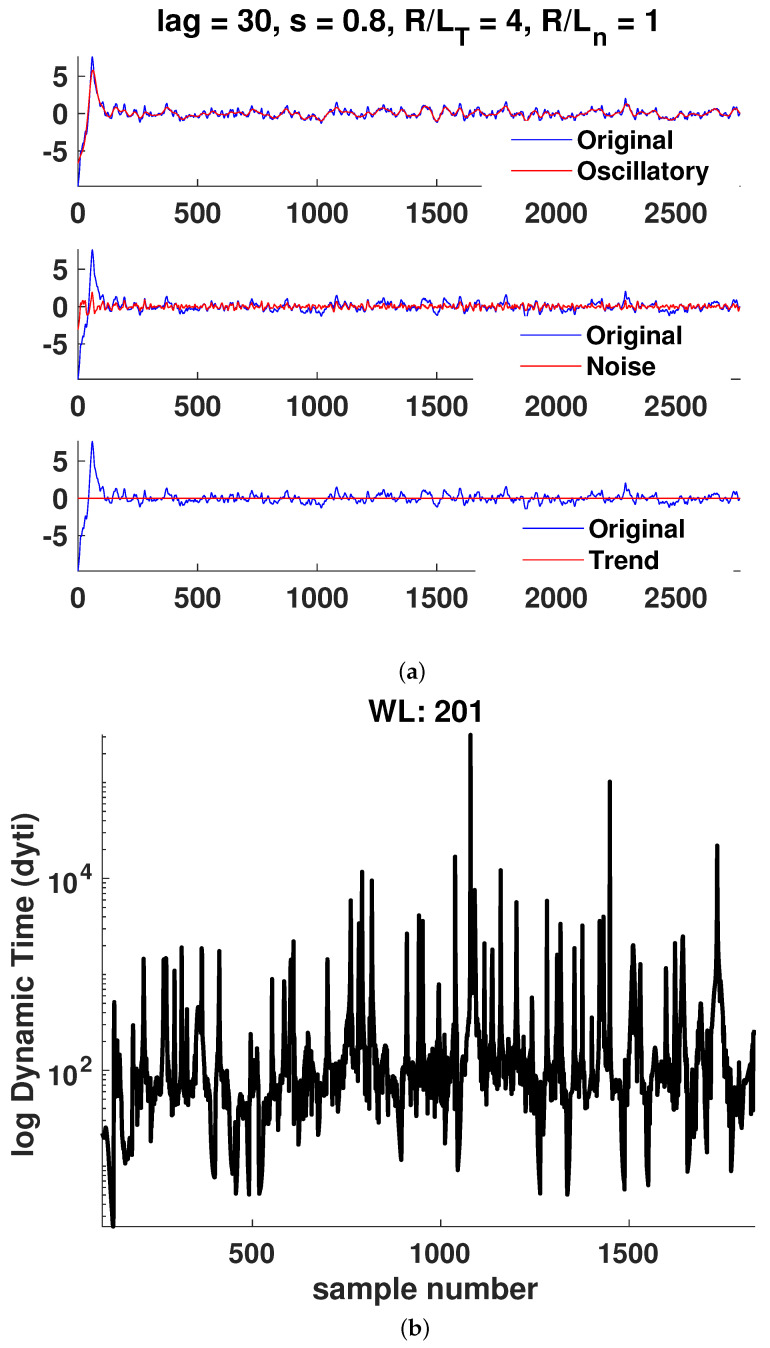
Case 6: (**a**) Hierarchical clustering results and (**b**) dynamic time calculations for WL=201. Here, nc is the maximal number of clusters, as defined in Section 3.

**Table 1 entropy-25-00942-t001:** The Hurst exponent and the β for all 8 realizations of the parameters *s*, $R/L_{T}$, and $R/L_{n}$ of the full time series.

$\left(s,R/L_{T},R/L_{n}\right)$	*H*	β=2H−1
$\left(0.5,1,0\right)$	0.7039	0.4078
$\left(0.5,4,0\right)$	0.5687	0.1374
$\left(0.5,1,1\right)$	0.6937	0.3874
$\left(0.5,4,1\right)$	0.6316	0.2632
$\left(0.8,1,0\right)$	0.6954	0.3908
$\left(0.8,4,0\right)$	0.5851	0.1702
$\left(0.8,1,1\right)$	0.5993	0.1986
$\left(0.8,4,1\right)$	0.5859	0.1718

**Table 2 entropy-25-00942-t002:** Maximum value of information length and Hurst exponent as a function of the parameter triplet *s*, $R/L_{T}$, and $R/L_{n}$.

CaseNumber	*s*	$R/L_{T}$	$R/L_{n}$	max(L)	*H*
1	0.5	1	0	41.3	0.7039
2	0.5	4	0	18.4	0.5687
3	0.5	4	1	58.0	0.6316
4	0.8	1	0	55.8	0.6954
5	0.8	4	0	43.4	0.5851
6	0.8	4	1	46.3	0.5859

**Table 3 entropy-25-00942-t003:** The signal-to-noise ratio in dB and the variance of the oscillatory and noise parts.

CaseNumber	(*s*, $R/L_{T}$, $R/L_{n}$)	SNR (dB)	VAR (Oscillatory)	VAR (Noise)
1	(0.5, 1, 0)	6.9	0.64	0.13
2	(0.5, 4, 0)	20.1	0.92	0.01
3	(0.5, 4, 1)	8.9	0.76	0.10
4	(0.8, 1, 0)	13.4	0.81	0.04
5	(0.8, 4, 0)	3.8	0.52	0.22
6	(0.8, 4, 1)	9.4	0.78	0.09

## Data Availability

The data that support the findings of this study are available from the corresponding author upon reasonable request.

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
