# Peer review of "Statistical Analysis of Plasma Dynamics in Gyrokinetic Simulations of Stellarator Turbulence"

_entropy, 2023, doi:10.3390/e25060942_

Round 1

Reviewer 1 Report

In this manuscript, a new geometrical method is proposed for assessing stochastic processes dominated by ITG-driven turbulent transport. By extracting the informative component from the time series, some statistical physical quantities such as the information length and the dynamic time can be calculated in addition to the Hurst exponent.

The proposed method itself is interesting and useful, however, the obtained results are not fully clear from the mathematical/physical view point. This paper is worthwhile for the publication in “Entropy” after a few revisions. Detailed comments on the manuscript are listed in the attached file.

English language is enough quality.

Author Response

We thank the referee for the constructive comments and valuable recommendations. The revised manuscript has been modified in accordance with the Reviewers suggestions and remarks, which have significantly improved the clarity and readability of the paper. All changes are marked with bold text.

Reviewer 2 Report

For testing of their approach, the Authors used data from numerical simulations of the stellarator plasma. The discussion is very convincing that the method based on the information theory can indeed be used to quantify plasma transport.

The manuscript has only one shortcoming. It would be nice to know if the theoretic/numerical predictions can be meaningfully compared with experimental data or if a realistic experimental setup yields enough data to even apply the proposed approach.

To conclude, the paper proposes an interesting new theoretical/numerical tool for quantifying plasma turbulence. Therefore, in my opinion the paper could be published in Entropy.

The quality of English is quite good. There are few minor errors scattered across the manuscript.

Author Response

(The authors gave the same response as above.)

Reviewer 3 Report

The work presented in this manuscript follows a well established vein. The authors produce several signals, in the form of time series. In this case the signals are those of fluxes produced by numerical simulations of electrostatic turbulence in a stellarator. Then, they extract features from the signals using statistical tools. In this case Singular Spectrum Analysis and Information Length. 

I have no serious complains. In my opinion the study has its motivations and is written correctly, therefore it may be published. 

I have just a few minor suggestions. 

-Line 50: the full stop appears before the references [18,19].

-At line 104 it is written "theta" rather than the Greek letter

-Line 127: I could not spot where the velocity coordinates v,w were defined, nor it is explained why 3 spatial coordinates and just 2 velocity coordinates are employed

-Line 131 and 142,142. The normalization of the quantities employed is explained at lines 142,143 whereas the (normalized) values and numerical bounds are given around line 131. I suggest to reverse this order.

-Lines 154,155. Chaotic and stochastic are not truly synonymous, therefore it is not appropriate using them interchangeably in the same sentence. 

-Line 223. A reference to the Hurst exponent would be advisable.

-Throughout the text the authors keep hopping between the s coordinate (0.5,0.8) and the x coordinate (0.4,0.7). I suggest introducing both, but sticking to just one of them. 

-The authors employ two reference values for the normalized temperature and density inverse scale lengths: 1 and 4. The first value corresponds to a scenario where microturbulence is stable (perhaps marginally stable), whereas the second scenario corresponds to fully destabilized turbulence. This is clear from the text but it is explicitly discussed only from lines 310 onwards. Again, in my opinion it is a better practice anticipating this discussion to the point where the parameters Ln, LT are introduced the first time. 

-There is a very long discussion about the Information length, from line 272 to 309, including figure 4-11. 

The statistical measure employed suffers from conflicting requirements: the more data one uses for building accurated PDFs, the coarser is the dynamics of the PDFs themselves that can be tracked.  

The authors carry out an extensive sensitivity study to this regard but, unfortunately, I did not find the whole discussion very clear. Ultimately, I guess that the authors identified an optimal choice of parameters to be used in the subsequent analysis but I am not even totally sure.  I invite the authors to reconsider this section. In detail, I invite them to consider the following comments:

* As far as I understand, the important parameter is WL. Its optimal value is inferred (if I understand correctly) from Figure 4, but it is not clear how: the authors write, at line 305: "convergent results are expected for WL = 201". Expected on the basis of what ? An explanation is needed.    

* The sentence above is found at page 17. It is referencing to figure 4, which is at page 10. In the middle there are 6 pages of figures produced using the parameter WL = 201 which is yet unjustified. I invite the authors to consider whether all figures are necessary (in particular I found very scarcely informative the content of figure 5), if could be moved to an appendix, or to reformat the section in such a way that figures 5-11 are located far away from figure 4, when the content of this latter figure has been properly discussed. 

Author Response

(The authors gave the same response as above.)

Round 2

Reviewer 1 Report

The authors have made appropriate revisions to their manuscript in accordance with the referee's comments. The revised manuscript is worthy of publication in the Journal of Entropy.

Reviewer 3 Report

The authors have satisfactorily replied to my comments/queries. In my opinion the manuscript may be published.